# In Vitro Growth-Inhibitory Synergistic Effect of Zinc Pyrithione in Combination with Gentamicin against Bacterial Skin Pathogens of Livestock

**DOI:** 10.3390/antibiotics11070960

**Published:** 2022-07-17

**Authors:** Lucie Mala, Klara Lalouckova, Eva Skrivanova, Marketa Houdkova, Marie Strakova, Ladislav Kokoska

**Affiliations:** 1Department of Microbiology, Nutrition and Dietetics, Faculty of Agrobiology, Food and Natural Resources, Czech University of Life Sciences Prague, Kamycka 129, 165 00 Prague, Czech Republic; malalucie@af.czu.cz (L.M.); lalouckova@af.czu.cz (K.L.); skrivanovae@af.czu.cz (E.S.); 2Department of Nutritional Physiology and Animal Product Quality, Institute of Animal Science, Pratelstvi 815, 104 00 Prague, Czech Republic; 3Department of Crop Science and Agroforestry, Faculty of Tropical AgriSciences, Czech University of Life Sciences Prague, Kamycka 129, 165 00 Prague, Czech Republic; houdkovam@ftz.czu.cz (M.H.); netopilova@ftz.czu.cz (M.S.)

**Keywords:** antimicrobial interaction, checkerboard microdilution method, fractional inhibitory concentration, zinc pyrithione, gentamicin

## Abstract

Bacterial skin diseases of livestock could be a serious global threat, especially in association with overcoming bacterial resistance. Combinatory action of antimicrobial agents proves to be an effective strategy to overcome the problem of increasing antibiotic resistance of microorganisms. In this study, the in vitro combined effect of zinc pyrithione with gentamicin against bacterial skin pathogens of livestock (*Staphylococcus aureus*, *Streptococcus agalactiae*, and *Streptococcus dysgalactiae*) was evaluated according to the sum of fractional inhibitory concentration indices (FICI) obtained by checkerboard method. The results showed that a combination of zinc pyrithione with gentamicin produced a strong synergistic effect (*p* < 0.001) against all tested streptococcal strains (with FICI values ranging from 0.20 to 0.42). Compared to that, only three out of eight *S. aureus* strains were highly susceptible to the combination of antimicrobial agents at single concentration (0.25 µg/mL) of zinc pyrithione with range of FICI 0.35–0.43. These findings suggest that interference between agents tested in this study can be used for the development of future veterinary pharmaceutical preparations for the treatment of bacterial skin infections of livestock.

## 1. Introduction

Besides various skin conditions of livestock caused by infectious agents such as parasites and viruses, the ulcers and wounds infected by bacteria and fungi are causing serious problems in animal production [1]. For example, foot rot, which is a hoof infection commonly found in sheep, goats, and cattle after interdigital skin injury, causes a significant financial impact, associated with the lost performance, preventive measures, and antibiotic treatment of affected animals; and estimated to be GBP 24 to GBP 80 million in annual losses within the United Kingdom alone [2]. The primary causes of delayed healing as well as the infections in acute and chronic wounds are *Staphylococcus aureus* [3] and beta-hemolytic streptococci such as *Streptococcus agalactiae* and *Str. dysgalactiae*. Other commonly isolated pathogens belong mainly to *Staphylococcus* spp., and *Streptococcus* spp. [4]. Bacterial skin infections of livestock are currently treated by using conventional antibiotics, such as amoxicillin, gentamicin (topical use), and penicillin G [5,6]. However, antibiotic treatment is accompanied by various disadvantages, including a low cure rate, serious side effects, and relatively high costs [7]. In addition, when the withdrawal period of drugs is not respected, the presence of antibiotic residues in animal products could occur [8]. Moreover, staphylococcal and streptococcal pathogens causing bacterial skin infections are of serious concern, mainly due to their increasing antibacterial resistance and potential transmission to other animals and the environment [4,9]. Specifically, epidemiology of *S. aureus* in livestock has gained interest in recent years because of the emergence of some clonal lineages associated with animals and their increasingly evidenced zoonotic potential [10]. Under these circumstances, there is a growing need for the development of alternatives to antibiotics in the prevention and treatment of bacterial skin infections [11].

One such approach is the utilization of plant-derived products that have the potential to be used as an alternative or a complement to antibiotics [12]. In the area of antibacterial plant-derived veterinary preparations, the broad spectrum of phytochemicals and their complex mixtures (e.g., extracts and essential oils) are nowadays used in various forms, such as poultices, ointments, soaps, and shampoos, to treat or prevent various bacterial skin diseases [13,14]. For example, Weaver Anti-dandruff Shampoo (Weaver Leather; USA) formulated with a combination of *Aloe vera*, essential oil from *Melaleuca alternifolia*, and zinc pyrithione, known for its ability to prevent and relieve itchy and flaky hide. Zinc pyrithione is an active antibacterial ingredient incorporated in an extensive range of topically applied commercial products for humans [15]. In veterinary medicine, it can be found only as a component of shampoos used to heal and protect skin and to treat and prevent dandruff [16]. Zinc pyrithione is a coordination complex of zinc ion and metal-binding pharmacophore pyrithione. Although chemists first synthesized both compounds, pyrithione has subsequently been identified as a constituent of the Chinese medicinal plant *Polyalthea nemoralis* as well as a decomposition product of sulphur-containing pyridine *N*-oxides in freshly disrupted plant tissue of the *Allium stipitatum* bulb. Therefore, zinc pyrithione can be classified as a synthetic analog of phytochemicals [17]. In vitro growth inhibitory effect of zinc pyrithione was described, for example, in the study of Blanchard et al. [18]. Specifically, it was shown to reduce levels of biofilm-associated *S. aureus* with minimum inhibitory concentration (MIC) 1 µg/mL. Based on the results of previous in vitro and in vivo studies focused on the antimicrobial activity of zinc pyrithione, a potential use of this substance in wound healing in livestock could be possible [18,19].

Another promising strategy to combat the rising problem of resistance is a combinatory antimicrobial therapy. Specifically, synergistic interaction between two agents, in which one agent enhances the effect of the other and together they act more efficiently than as individual agents, are well known [20,21]. Synergism against bacterial skin pathogens such as *S. aureus* or *Streptococcus* spp. can frequently be achieved with combinations of penicillins, cephalosporins and aminoglycosides [22,23]. The combination of clavulanic acid, an inhibitor of β-lactamase enzymes, with β-lactam antibiotics (e.g., amoxicillin or ticarcillin), is the best-known example of preparation used in medical veterinary practice [24]. In contrast, preparation based on synergistically acting plant products or their derivatives efficient against bacterial skin pathogens of farm animals is still not commercially available [25]. Based on the results of our preliminary screenings performed as several combinations of zinc pyrithione with different antibiotics (erythromycin, clindamycin, gentamicin, oxacillin, penicillin, and vancomycin) against *S. aureus* of the American Type Culture Collection (ATCC) 29213, the combination of zinc pyrithione with gentamicin produced the lowest fractional inhibitory concentration (FIC) value 0.61 (unpublished data). Taking into account these laboratory results and into the synergistic potential of gentamicin against bacterial strains tested in this study [26,27], a theoretical growth-inhibitory synergistic effect of zinc pyrithione in combination with gentamicin could be hypothesized. Therefore, in the present study, we evaluated the in vitro synergistic effect of these substances through the checkerboard microdilution method against various *S. aureus* strains and strains of the genus *Streptococcus* spp. related to skin and soft tissue infections of livestock.

## 2. Results

In this study, zinc pyrithione demonstrated significant synergistic antistreptococcal and antistaphylococcal activity with gentamicin (*p* < 0.001). Compared to *S. aureus* strains, *Str. agalactiae* and *Str. dysgalactiae* were more sensitive (*p* < 0.001) to the combination of the zinc pyrithione/gentamicin. The strongest synergy (ΣFIC 0.20) was obtained against the strain of *Str. agalactiae* isolated from bovine udder infection (DSM 6784) at a zinc pyrithione concentration of 0.031 µg/mL, when a 6.39-fold gentamicin MIC decrease was achieved (from 2.11 to 0.33 µg/mL). Moreover, all other tested strains of *Str. agalactiae* and *Str. dysgalactiae* were susceptible to the zinc pyrithione/gentamicin combination which showed many synergistic interactions with a range of ΣFIC 0.20–0.42. Although no interaction was found in most combinations of tested substances against *S. aureus*, three out of eight *S. aureus* strains were highly susceptible to the combination of antimicrobial agents at single concentration (0.25 µg/mL) of zinc pyrithione with range of ΣFIC 0.35–0.43. None of the tested combinations exerted antagonistic effect. The detailed susceptibilities of all tested bacterial strains to the gentamicin, either alone or in the presence of zinc pyrithione, are presented in Table 1, Table 2 and Table 3.

The combination profiles of the most sensitive bacterial strains are presented graphically in form of isobologram curves (Figure 1 and Figure 2), which represents the results of the checkerboard assay and the FICI values, whereas the axes of each isobologram are the dose-axes of the individual agents. If synergy occurs, the curve becomes “concave”—an inward curve. The resulting isobolograms confirmed the synergistic effect of the zinc pyrithione/gentamicin combinations against *Str. agalactiae* strains, where synergy was observed for four ratios in the isobolograms, for three ratios in *Str. dysgalactiae* strains, and for one ratio in *S. aureus* strains. The isobole curve of synergy is shown as a red line (dashed line), with a distinct concave shape.

## 3. Discussion

Regarding antibacterial activity of gentamicin against *S. aureus* strains, our findings show no remarkable differences between values of MICs observed in this study and sensitivity of *S. aureus* ATCC 29213 (MICs 0.12–1 µg/mL) interpreted by Clinical and Laboratory Standards Institute [28]. Similarly, our results are comparable with previous data of Hu et al. [22], who reported MICs values of gentamicin ranging from 0.5 to 4 µg/mL for 101 clinical isolates of *S. aureus* as well as with the results of Paduszynska et al. [29], who determined its antimicrobial effect against *S. aureus* ATCC 25923 (MIC 0.25 µg/mL). According to the Manual of Clinical Microbiology, susceptibility to gentamicin is generally defined as a MICs ≤ 4 µg/mL [30]. Based on this, all strains of *Str. agalactiae* and *Str. dysgalactiae* assayed in this study could be considered as a sensitive to gentamicin. In contrast, literature reports significantly lower susceptibility of streptococcal strains. For example, Lin et al. [31] reported MICs values of gentamicin ranging from 32 to 128 µg/mL toward 42 clinical isolates of *Str. agalactiae* using microdilution broth method. Similarly, Oh et al. [32] determined MICs values of gentamicin ranging from 8 to 16 µg/mL against clinical isolate of Str. dysgalactiae subspecies equisimilis. In addition, Moreno et al. [33] reported MIC = 8 mg/L for standard strain of *Str. agalactiae* ATCC 13813. The variations between of our results and previously published data can be explained by different gentamicin susceptibility of various streptococcal strains tested. Although antibacterial activity of zinc pyrithione has been proven, data on its in vitro growth-inhibitory effects against staphylococcal and streptococcal strains are limited. Out of the bacteria tested herein, only Blanchard et al. [18] reported antimicrobial effects of zinc pyrithione against *S. aureus* UAMS-1 (MIC = 1 µg/mL), which is in line with our results, where values of MICs did not exceed 1 µg/mL. To the best of our knowledge, this is the first report on the synergistic effect of the combination of zinc pyrithione and gentamicin. However, a few studies dealing with combinatory effect of zinc pyrithione with other antibiotics are known. For example, the previously mentioned research of Blanchard et al. [18], where the zinc pyrithione/sulfadiazine combination exhibited additive antimicrobial effects on biofilm-associated *S. aureus* UAMS-1 (FIC = 0.52). Based on this, the present study provides new data on in vitro growth-inhibitory synergistic effect of zinc pyrithione and gentamicin combination towards bacterial skin pathogens of livestock. Although our data show clearly antibacterial efficacy of this combination, optimization of MICs by assessing all possible combinations of zinc pyrithione and gentamicin would provide more valid results in economic terms. A giant checkerboard method, which enables determination of MICs with greater accuracy and produces better estimates of the parameters used to measure the interaction between antibiotics, can be option for identification of optimal MIC values. However, this method requires substantially more labor and laboratory materials [34].

It has previously been documented that gentamicin binds to the 30S ribosomal subunit, which leads to a misreading of the messenger ribonucleic acid (m-RNA), thereby inducing inhibition of protein biosynthesis, and causing the bacterial cell death [35]. The m-RNA is created during the process of transcription in the presence of RNA polymerase [36]. Metal ion cofactors are known to affect substantially the biological properties of this enzyme [37]. In particular, Mg^2+^ is an important cofactor of RNA polymerases [38], which contributes to present structures of large and small ribosomal subunits [39]. Therefore, it is possible to assume that zinc pyrithione, which is known to chelate metal ions [40], is creating complexes with Mg^2+^, thereby making it unavailable to the process of protein biosynthesis (Figure 3). Moreover, it has previously been demonstrated that activity of aminoglycosides is reduced in the presence of divalent cations, such as Mg^2+^ [41]. The ability of Mg^2+^ to form complexes with zinc pyrithione as well as to reduce activity of gentamicin may therefore significantly contribute to the synergistic antibacterial action of both compounds observed in this study. In addition, the physical properties of antimicrobials also greatly affect their antibacterial activity. Gentamicin is a highly polar hydrophilic substance [42] which passes through the bacterial membrane in an oxygen-dependent active transport [43]. Based on this, it can also be hypothesized that zinc pyrithione, which is membrane active [44] and forms the disaggregation of the phospholipid head structure at the outer membrane [40], can facilitate the passage of gentamicin into the bacterial cells (Figure 4). The synergistic activity of zinc pyrithione in combination with gentamicin can therefore be explain by all above-mentioned mechanisms and actions. However, further research focused on a better understanding of antimicrobial interactions between zinc pyrithione and gentamicin is warranted.

The safety of new antimicrobials is essential to prove their usage in veterinary practice [45]. According to the National Registration Authority for Agricultural and Veterinary Chemicals, dermal exposure of zinc pyrithione is practically non-toxic to mammals with a median lethal dose to albino rabbits > 2000 mg/kg [46]. Its permeation to the skin is very low, typically representing less than 0.05% of the initial applied dose [47]. For example, a 0.1% solution of the zinc pyrithione soap injected intracutaneously into depilated guinea pig skin at an initial dose of 0.05 mL and nine subsequent doses of 0.1 mL on alternate weekdays, did not cause any evidence of sensitization [48]. Another study showed that it is very poorly (<10% of dose) absorbed through skin of pigs [46]. In addition, lifetime studies using mice and rats demonstrated no evidence of carcinogenic potential from dermal doses up to 100 mg/kg/day. Similarly, no teratogenic effects have been observed in rabbits topically treated with shampoo containing up to 50 mg/kg/day of zinc pyrithione. Finally, the local use of its levels up to 15 and 100 mg/kg/day also did not confirm the reproductive effects in rats and rabbits, respectively [48]. The above-mentioned data suggest a low toxicological risk of zinc pyrithione administration through the skin route which could support its use in combination with gentamicin, an antibiotic frequently used in animals, as a safe veterinary product for treatment of skin infections in livestock. Although our in vitro data cannot be directly interpreted for in vivo use, 8–10 times the MIC of the substances tested, when the optimal bactericidal effect occurs, could be recommended for further research [49] However, the MIC values gains clinical significance only in relation to pharmacokinetic parameters that describe the fate of the drug in the host organism; therefore, the next step in research should be the study of these parameters [50].

## 4. Materials and Methods

### 4.1. Chemicals

Zinc pyrithione and gentamicin were purchased from Sigma-Aldrich (Prague, Czechia). Dimethyl sulfoxide (Sigma-Aldrich) was used to prepare the stock solution of zinc pyrithione, whereas gentamicin was diluted in distilled water.

### 4.2. Bacterial Strains and Growth Media

In this study, eight *S. aureus* strains and four strains of the genus *Streptococcus* spp., were used. Standard strain of the *S. aureus* ATCC 29213 was purchased from Oxoid (Basingstoke, UK) on ready-to-use bacteriological Culti-Loops. The clinical isolates (SA1, SA3) were obtained from the Motol University Hospital (Prague, Czechia) and identified using matrix-assisted laser desorption/ionization time-of-flight mass spectrometry (MALDI-TOF MS), as described previously by Rondevaldova et al. [51]. *S. aureus* CCM 885, CCM 2022, CCM 2773, CCM 4516; and *Str. agalactiae* CCM 6187, were purchased from the Czech Collection of Microorganisms (Brno, Czechia). *S. aureus* DSM 6732; *Str. agalactiae* DSM 6784; and *Str. dysgalactiae* DSM 2734, DSM 20662; were obtained from the German Resource Centre for Biological Material (Braunschweig, Germany). Cation-adjusted Mueller-Hinton Broth (Oxoid) was used as the cultivation and assay medium for all tested strains of *S. aureus*. CCM and DSM strains were cultured in Tryptone Soya Broth (Oxoid), which was enriched by 3 g/L yeast extract (Sigma-Aldrich) in the case of DSM strains. All used growth media were equilibrated to pH 7.6 with a Trizma base (Sigma-Aldrich).

### 4.3. Evaluation of Minimum Inhibitory Concentrations and Synergistic Combinatory Effect

Individual MICs of gentamicin and zinc pyrithione were determined by the broth microdilution method as described by the Clinical and Laboratory Standards Institute [52], as modified according to Cos et al. [53] in their recommendations proposed for the effective assessment of the anti-infective potential of natural products. Antistaphylococcal and antistreptococcal combinatory effect of zinc pyrithione/gentamicin were evaluated by the checkerboard method based on the FICI, as described in the Clinical Microbiology Procedures Handbook [54]. The determination of MICs of gentamicin and zinc pyrithione as well as gentamicin/zinc pyrithione combinatory effect evaluation by FICI were performed in 96-well microtiter plates.

In the combinations, eight two-fold serial dilutions of gentamicin from horizontal rows of the microtiter plate were subsequently crossdiluted vertically by eight two-fold serial dilutions of the zinc pyrithione. Microplates so arranged can be used to screen 64 different combinations of concentrations. The initial concentrations used in the combinations for zinc pyrithione were 4 µg/mL. The only exception was strain *Str. agalactiae* CCM 6187, where it was 1 µg/mL. Compared to that, various starting concentrations were used depending on the staphylococcal and streptococcal strain’s susceptibility to gentamicin.

Plates were inoculated by bacterial suspension at a final density 5 × 10^5^ CFU/mL using the McFarland scale and incubated at 37 °C for 24 h under aerobic conditions. The bacterial growth was then assessed as the turbidity determined by an Infinite 200 PRO microplate reader (Tecan, Männedorf, Switzerland) at 405 nm according to Cos et al. [53]. Gentamicin was used as a positive antibiotic control for verification of susceptibility of bacterial strains in the broth medium. A drug-free bacterial culture served as the negative control. The MICs were expressed as the lowest concentrations that inhibited bacterial growth by ≥80% compared with that of the agent-free growth control. All substances and their combinations were tested in three independent experiments, each carried out in triplicate; MIC values presented in this paper are average values.

The combinatory effect of zinc pyrithione with gentamicin was determined based on the value of ΣFIC. For the combination of agent A (gentamicin) and agent B (zinc pyrithione), the ΣFIC was calculated according to the following equation: ΣFIC = FIC_A_ + FIC_B_, where FIC_A_ = MIC_A (in combination with B)_/MIC_B (alone)_, and FIC_B_ = MIC_B (in combination with A)_/MIC_B (alone)_ and evaluated according to Odds [55]. The ΣFIC index was interpreted as follows: synergistic interaction if ΣFIC ≤ 0.5; indifference if ΣFIC > 0.5–4, and antagonism if ΣFIC > 4.

### 4.4. Statistical Analysis

The statistically significant differences (*p* < 0.001) between FIC values of the individual strains tested (in terms of the same concentration of zinc pyrithione within Table 1, Table 2 and Table 3) were analysed using one-way analysis of variance (ANOVA) with the general linear model procedure, followed by Tukey’s HSD test, in SAS software (version 9.1, 2003, Cary, NC, USA). The main effect was the individual bacterial strains.

## 5. Conclusions

In this study, zinc pyrithione in combination with gentamicin produced synergistic antibacterial effect against bacterial skin pathogens of livestock, whereas the best results were obtained against streptococcal strains. To the best of our knowledge, this is the first report on interactions between zinc pyrithione and gentamicin against *S. aureus*, *Str. agalactiae* and *Str. dysgalactiae* strains. Due to the considerable antimicrobial activity as well as the presumable safety of zinc pyrithione, it can be assumed that the combinatory actions of zinc pyrithione/gentamicin could be potentially used in the development of future veterinary pharmaceutical preparations for treatment of bacterial skin infections. These combinations could decrease the minimum effective doses of the agents, thus reducing their possible adverse effects and livestock treatment costs. Simultaneously, the synergistic antimicrobial effect of the tested substances could become an important mediator to overcome bacterial resistance. However, the above-mentioned assumption is based on the results of in vitro test only and in vivo studies focused on the efficacy and toxicity of these compounds in animal models will be needed before their consideration to be used in veterinary medicine.

## Figures and Tables

**Figure 1 antibiotics-11-00960-f001:**
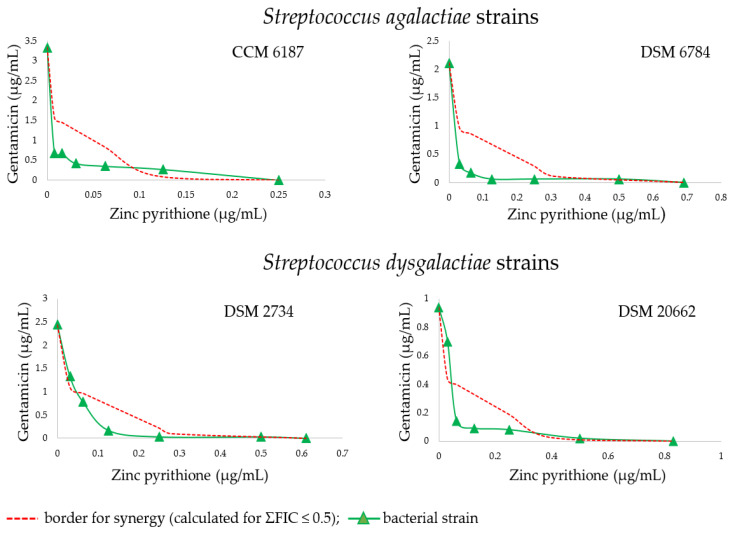
Isobolograms of the interactions between gentamicin and zinc pyrithione binary combinations against *Streptococcus agalactiae* and Streptococcus dysgalactiae strains. Synergy (ΣFIC ≤ 0.5).

**Figure 2 antibiotics-11-00960-f002:**
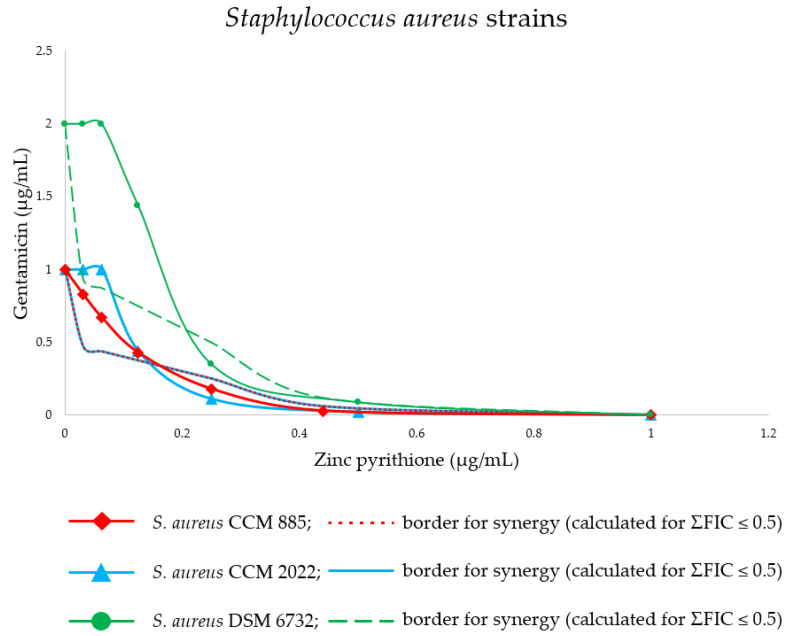
Isobologram of the interactions between gentamicin and zinc pyrithione binary combinations against *Staphylococcus aureus* strains. Synergy (ΣFIC ≤ 0.5).

**Figure 3 antibiotics-11-00960-f003:**
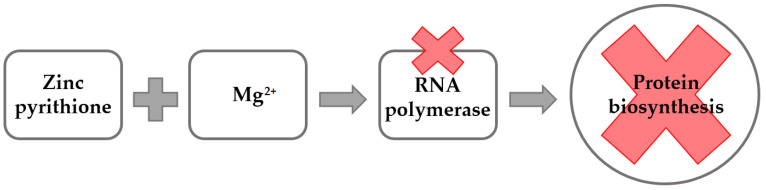
A diagram showing the effect of complex of zinc pyrithione with Mg^2+^ on the process of protein biosynthesis.

**Figure 4 antibiotics-11-00960-f004:**
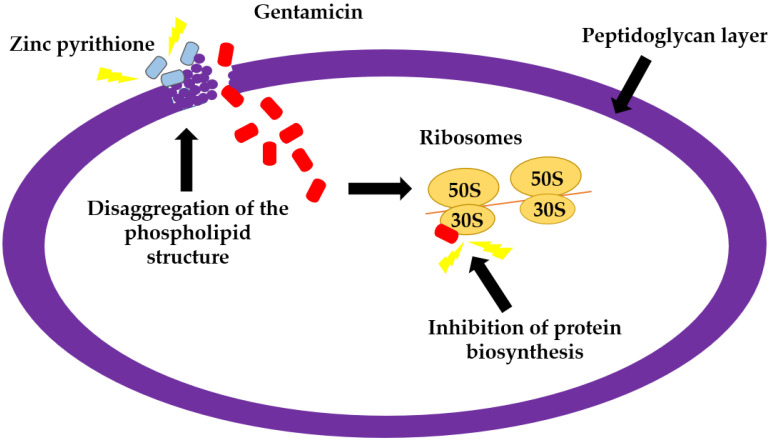
The penetration of zinc pyrithione and gentamicin to the cell membrane.

**Table 1 antibiotics-11-00960-t001:** In vitro inhibitory activity of zinc pyrithione/gentamicin combinations against *Streptococcus agalactiae* strains.

*Str.a.* strain		Alone MICs (µg/mL)			GEN at concentration indicated in MIC column in combination with listed ZnP concentrations (µg/mL)													
		GEN	ZnP		+ZnP 0.125			+ZnP 0.063			+ ZnP 0.031			+ZnP 0.016			+ZnP 0.008	
					MIC	ΣFIC		MIC	ΣFIC		MIC	ΣFIC		MIC	ΣFIC		MIC	ΣFIC
6187 *		3.33	0.25		0.27	0.58 ^b^		0.35	**0.36** ^cd^		0.42	**0.25** ^d^		0.67	**0.26**		0.67	**0.23**
					+ZnP 0.50			+ZnP 0.25			+ZnP 0.125			+ZnP 0.063			+ZnP 0.031	
		GEN	ZnP		MIC	ΣFIC		MIC	ΣFIC		MIC	ΣFIC		MIC	ΣFIC		MIC	ΣFIC
6784 **		2.11	0.69		0.06	0.75 ^a^		0.06	**0.39** ^b^		0.06	**0.26 ^c^**		0.17	**0.25 ^cd^**		0.33	**0.20** ^d^

Bold values: synergy (ΣFIC ≤ 0.5). * CCM, Czech Collection of Microorganisms. ** DSM, German Collection of Microorganisms and Cell Cultures GmbH. *Str.a*., *Streptococcus agalactiae*; GEN, gentamicin; ZnP, zinc pyrithione; MIC, minimum inhibitory concentration (expressed as an average of three independent experiments performed in triplicate); ΣFIC, sum of fractional inhibitory concentrations. Different bold small letters (^a–d^) represent significant differences (*p* < 0.001) between FICI of the individual strains (in terms of single concentration of zinc pyrithione) within Table 1, Table 2 and Table 3 by one-way analysis of variance (ANOVA) followed by Tukey’s HSD test.

**Table 2 antibiotics-11-00960-t002:** In vitro inhibitory activity of zinc pyrithione/gentamicin combinations against *Streptococcus dysgalactiae* strains.

*Str.d.* strain		Alone MICs (µg/mL)			GEN at concentration indicated in MIC column in combination with listed ZnP concentrations (µg/mL)													
		GEN	ZnP		+ZnP 0.50			+ZnP 0.25			+ZnP 0.125			+ZnP 0.063			+ZnP 0.031	
					MIC	ΣFIC		MIC	ΣFIC		MIC	ΣFIC		MIC	ΣFIC		MIC	ΣFIC
2734 *		2.44	0.61		0.03	0.83 ^a^		0.03	**0.42** ^b^		0.17	**0.27** ^c^		0.78	**0.39** ^cd^		1.33	0.59 ^c^
20662 *		0.94	0.83		0.02	0.62 ^b^		0.08	**0.39** ^b^		0.09	**0.24** ^c^		0.14	**0.22** ^cd^		0.70	0.51 ^c^

Bold values: synergy (ΣFIC 0.5). * DSM, German Collection of Microorganisms and Cell Cultures GmbH. *Str.d*., *Streptococcus dysgalactiae*; GEN, gentamicin; ZnP, zinc pyrithione; MIC, minimum inhibitory concentration (expressed as an average of three independent experiments performed in triplicate); ΣFIC, sum of fractional inhibitory concentrations. Different bold small letters (^a–d^) represent significant differences (*p* < 0.001) between FICI of the individual strains (in terms of single concentration of zinc pyrithione) within Table 1, Table 2 and Table 3 by one-way analysis of variance (ANOVA) followed by Tukey’s HSD test.

**Table 3 antibiotics-11-00960-t003:** In vitro inhibitory activity of zinc pyrithione/gentamicin combinations against *Staphylococcus aureus* strains.

*S.a.* strain		Alone MICs (µg/mL)			GEN at concentration indicated in MIC column in combination with listed ZnP concentrations (µg/mL)													
		GEN	ZnP		+ZnP 0.50			+ZnP 0.25			+ZnP 0.125			+ZnP 0.063			+ZnP 0.031	
					MIC	ΣFIC		MIC	ΣFIC		MIC	ΣFIC		MIC	ΣFIC		MIC	ΣFIC
SA1 *		0.67	0.50		0.02	1.03 ^a^		0.03	0.55 ^a^		0.39	0.83 ^a^		1.12	1.80 ^a^		1.56	2.39 ^a^
SA3 *		1.00	0.50		0.02	1.02 ^a^		0.03	0.53 ^a^		0.5	0.75 ^a^		0.89	1.02 ^bc^		0.94	1.00 ^bc^
29213 **		1.00	0.50		0.02	1.02 ^a^		0.11	0.61 ^a^		0.63	0.88 ^a^		1.21	1.33 ^b^		1.33	1.39 ^b^
885 ***		1.00	1.00		0.03	0.53 ^bc^		0.18	**0.43** ^b^		0.43	0.54 ^b^		0.67	0.73 ^c^		0.83	0.86 ^bc^
2022 ***		1.00	1.00		0.02	0.52 ^c^		0.11	**0.35** ^bc^		0.44	0.57 ^b^		1.00	1.06 ^bc^		1.00	1.03 ^bc^
2773 ***		2.00	0.50		0.02	1.01 ^a^		0.09	0.55 ^a^		0.54	0.52 ^b^		1.58	0.92 ^bc^		1.67	0.90 ^bc^
4516 ***		1.00	0.50		0.02	1.02 ^a^		0.12	0.62 ^a^		0.64	0.89 ^a^		1.33	1.46 ^b^		1.61	1.67 ^b^
6732 ****		2.00	1.00		0.09	0.55 ^bc^		0.35	**0.42** ^b^		1.44	0.85 ^b^		2.00	1.06 ^bc^		2.00	1.03 ^bc^

Bold values: synergy (ΣFIC ≤ 0.5). * Clinical isolate. ** ATCC, American Type Culture Collection. *** CCM, Czech Collection of Microorganisms. **** DSM, German Collection of Microorganisms and Cell Cultures GmbH. *S.a*., *Staphylococcus aureus*; GEN, gentamicin; ZnP, zinc pyrithione; MIC, minimum inhibitory concentration (expressed as an average of three independent experiments performed in triplicate); ΣFIC, sum of fractional inhibitory concentrations. Different bold small letters (^a–c^) represent significant differences (*p* < 0.001) between FICI of the individual strains (in terms of single concentration of zinc pyrithione) within Table 1, Table 2 and Table 3 by one-way analysis of variance (ANOVA) followed by Tukey’s HSD test.

## Data Availability

The data presented in this study are available in this article.

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
