# Peer review of "In Vitro Growth-Inhibitory Synergistic Effect of Zinc Pyrithione in Combination with Gentamicin against Bacterial Skin Pathogens of Livestock"

_antibiotics, 2022, doi:10.3390/antibiotics11070960_

Round 1

Reviewer 1 Report

I have evaluated the manuscript (Antibiotics-1777487) titled “In vitro growth-inhibitory synergistic effect of zinc pyrithione in combination with gentamicin against bacterial skin pathogens of livestock” by Kokoska and Co-workers, and the author has done an extensive research to develop future veterinary pharmaceutical preparations for the treatment of bacterial skin infections of livestock using combination therapy of zinc pyrithione with gentamicin. Excellent presentation of results in the manuscript and clearly describing the outcome.  I found the document interesting for the readers and follow the scope of the journal Antibiotics.

I would like to recommend the article could be published in Antibiotics, without any revision.

The authors could make the following minor changes.

1.     Introduction is lengthy. The author could some part of the introduction to the result and discussion section.

2.     The author could have discussed the rationale for the percentage of zinc pyrithione and gentamicin used for the study.  From the economical viewpoint, did the author use all possible combinations of zinc pyrithione and gentamicin to get optimal MIC?

3.     The author could have shown a pictorial presentation of penetration of zinc pyrithione and gentamicin to the cell membrane and a diagram showing a complex of Mg2+ to interrupt the process of protein bi- biosynthesis.

4.     The author could have mentioned about positive control or negative control used for the study.

5.     All the references should be in the same format.

(a)   Year in bold (ref. 5, 6, 21, 22, 29, 44, 46, 48, 50).

(b)  For pages use the same format for all (ref 5, 21, 29, 50 remove pp. for pages)

(c)   Use for a range of pages either 293-299 or 65–73 format for all.

Author Response

Itemized response to the reviewer's comments

Manuscript-ID: antibiotics-1777487

Reviewer: 1

Query: Introduction is lengthy. The author could some part of the introduction to the result and discussion section.

Response: We shortened the original version of the Introduction section from 765 words to 711 words. However, according to the Reviewer 2 other information has been added. For that reason, the Introduction section has 767 words (after a complete revision).

Query: The author could have discussed the rationale for the percentage of zinc pyrithione, and gentamicin used for the study. From the economical viewpoint, did the author use all possible combinations of zinc pyrithione and gentamicin to get optimal MIC?

Response: With aim to comment the results of experiments from economical viewpoint (MIC optimization), we added following text to the Discussion section: “Although our data show clearly antibacterial efficacy of this combination, optimization of MICs by assessing all possible combinations of zinc pyrithione and gentamicin would provide more valid results in economic terms. A giant checkerboard method, which enables determination of MICs with greater accuracy and produces better estimates of the parameters used to measure the interaction between antibiotics, can be option for identification of optimal MIC values. However, this method requires substantially more labor and laboratory materials [32].”

Following reference was added:

[32] Hsieh, M.H.; Yu, C.M.; Yu, V.L.; Chow, J.W. Synergy assessed by checkerboard. A critical analysis. Diagnostic Microbiology and Infectious Disease 1993, 16, 343-349 https://doi.org/10.1016/0732-8893(93)90087-n.

Query: The author could have shown a pictorial presentation of penetration of zinc pyrithione and gentamicin to the cell membrane and a diagram showing a complex of Mg2+ to interrupt the process of protein biosynthesis.

Response: We included a simplified diagram (Figure 3) showing the effect of complex of zinc pyrithione with Mg2+ on the process of protein biosynthesis and a pictorial presentation (Figure 4) of penetration of zinc pyrithione and gentamicin to the cell membrane, to the Discussion section.

Query: The author could have mentioned about positive control or negative control used for the study.

Response: We included information about positive and negative control to the Materials and Methods section. Following text was added: “Gentamicin was used as a positive antibiotic control for verification of susceptibility of bacterial strains in broth medium. A drug free bacterial culture served as the negative control.”

Query: All the references should be in the same format.

(a)   Year in bold (ref. 5, 6, 21, 22, 29, 44, 46, 48, 50).

(b)  For pages use the same format for all (ref 5, 21, 29, 50 remove pp. for pages)

(c)   Use for a range of pages either 293-299 or 65–73 format for all.

Response: (a),(b): In case of writing of reference, we followed Antibiotics guidelines for authors. Year in bold is required only in Journal Articles, and pp for pages is necessary in Books and Book Chapters (https://www.mdpi.com/journal/antibiotics/instructions#references).

       (c): We used the same format for all range of pages (65–73).

Reviewer 2 Report

Authors evaluated an innovative combination of zinc pyrithione and gentamicin against some of the major bacterial skin pathogen of livestock animals. The study is interesting since innovative strategies are needed to cope with the increasing problem of antibiotic resistance. The paper is well written and provides some interesting insight on the topic. I have only some comments in order to clarify some aspects before considering this manuscript for the publication.

In general, all Latin words (e.g. in vitro and in vivo) have to be listed in italic. I notice that there is no statistical analysis chapter, why did you not consider possible comparisons among treatments/strains and concentrations?

Abstract:

Substitute “combinatory” with “combined”.

Line 18: Use the italic for all Latin words “in vitro”.

Line 23: Provide the p-value for statistically significant differences.

Line 36: Is the economic impact due to antibiotic costs?

Line 46: Antibiotic residues should be avoided respecting the withdrawal period for each drug.

Line 72: Use the italic for in vitro.

Line 75: The same comment as above (in vitro and in vivo).

Line 77: Is zinc pyrithione extracted from plants or synthetically produced? Can it be considered as phytochemical?

Line 89: in vitro.

Line 92: The hypothesis of the study is lacking.

Results

Every time that authors refer to a statistically significant difference the p-value should be added between brackets.

Line 96: More sensitive compared to? Try to be more precise and provide the p-value

Line 109: Where are other graphs related to other bacterial strains tested?

Table 1-2-3: Please add the statistical differences using lowercase letters and p-values.

Table 3: Substitute the comma with the dot for the separation of decimal numbers. Please list all values with two decimal places.

Line 178: Causing the bacterial cell death.

Line 215: Even if the in vitro data can not be directly translated to in vivo, based on your results, which concentration of what molecules do you suggest for the treatment of skin bacterial infections?

Materials and methods

Line 256: Why did you measure at 405 nm of wavelength?

Statistical analysis is lacking, did you compare treatments and concentrations by statistical tests?

Author Response

Itemized response to the reviewer's comments

Manuscript-ID: antibiotics-1777487

Reviewer: 2

Query: In general, all Latin words (e.g. in vitro and in vivo) have to be listed in italic.

Response: In case of writing of Latin words, we followed MDPI guidelines for authors. According these guidlines, Latin terms (e.g. in vitro and in vivo) do not need to be highlighted or italicized (https://www.mdpi.com/authors/layout).

Query: I notice that there is no statistical analysis chapter, why did you not consider possible comparisons among treatments/strains and concentrations?

Response: We added following subsection 4.4. “Statistical analysis” to the Materials and Methods section: “The statistically significant differences (P < 0.001) between FIC values of the individual strains tested (in terms of the same concentration of zinc pyrithione within Tables 1, 2 and 3) were analysed using 2-way analysis of variance (ANOVA) with the general linear model procedure in SAS software (version 9.1, 2003, North Carolina, USA).

Query: Substitute “combinatory” with “combined”.

Response: The sentence was modified as follows: “In this study, the in vitro combined effect of zinc pyrithione with gentamicin against bacterial skin pathogens of livestock (Staphylococcus aureus, Streptococcus agalactiae, and Streptococcus dysgalactiae) was evaluated according to the sum of fractional inhibitory concentration indices (FICI) obtained by checkerboard method.”

Query: Line 18: Use the italic for all Latin words “in vitro”.

Response: In case of writing of Latin words, we followed MDPI guidelines for authors. According these guidlines, Latin terms (e.g. in vitro and in vivo) do not need to be highlighted or italicized (https://www.mdpi.com/authors/layout).

Query: Line 23: Provide the p-value for statistically significant differences.

Response: The p-value was provided. The sentence was modified as follows: “The results showed that combination of zinc pyrithione with gentamicin produced strong synergistic effect (P < 0.001) against all tested streptococcal strains (with FICI values ranging from 0.20 to 0.42).”

Query: Line 36: Is the economic impact due to antibiotic costs?

Response: With aim to clarify causes of economic impact, the text has been modified as follows: “For example, foot rot that is a hoof infection commonly developing in sheep, goats, and cattle after interdigital skin injury, causes a significant financial impact, associated with the lost performance, preventive measures, and antibiotic treatment of affected animals; and estimated to be £24 to £80 million in annual losses within the United Kingdom alone [2].”

Query: Line 46: Antibiotic residues should be avoided respecting the withdrawal period for each drug.

Response: With aim to specify the occurrence of antibiotic residues with regard to the withdrawal period of drugs, the text has been modified as follows “However, antibiotic treatment is accompanied by various disadvantages, including a low cure rate, serious side effects, and relatively high costs [7]. In addition, in case of not respecting withdrawal period of drugs, the presence of antibiotic residues in animal products could occur [8].”

Query: Line 72: Use the italic for in vitro.

Response: In case of writing of Latin words, we followed MDPI guidelines for authors. According these guidlines, Latin terms (e.g. in vitro and in vivo) do not need to be highlighted or italicized (https://www.mdpi.com/authors/layout).

Query: Line 75: The same comment as above (in vitro and in vivo).

Response: In case of writing of Latin words, we followed MDPI guidelines for authors. According these guidlines, Latin terms (e.g. in vitro and in vivo) do not need to be highlighted or italicized (https://www.mdpi.com/authors/layout).

Query: Line 77: Is zinc pyrithione extracted from plants or synthetically produced? Can it be considered as phytochemical?

Response: With aim to specify the origin of zinc pyrithione, we added following sentence to the Introduction section: “Therefore, zinc pyrithione can be classified as a synthetic analog of phytochemicals.”

Query: Line 89: in vitro.

Response: In case of writing of Latin words, we followed MDPI guidelines for authors. According these guidlines, Latin terms (e.g. in vitro and in vivo) do not need to be highlighted or italicized (https://www.mdpi.com/authors/layout).

Query: Line 92: The hypothesis of the study is lacking.

Response: The hypothesis of the study was added to the Introduction section: “Despite the existence of several studies dealing with combinatory effect of zinc pyrithione with other antibiotics [18], the combined effect of zinc pyrithione with aminoglycosides is still largely unknown.”

Query: Every time that authors refer to a statistically significant difference the p-value should be added between brackets.

Response: The p-value was added between brackets for statistically significant differences in main text as follows - Abstract section: “The results showed that combination of zinc pyrithione with gentamicin produced strong synergistic effect (P < 0.001) against all tested streptococcal strains (with FICI values ranging from 0.20 to 0.42).”; Result section: “In this study, zinc pyrithione demonstrated significant synergistic antistreptococcal and antistaphylococcal activity with gentamicin (P < 0.001).”, “Compared to S. aureus strains, Str. agalactiae and Str. dysgalactiae were more sensitive (P < 0.001) to the combination of the zinc pyrithione/gentamicin.”

Query: Line 96: More sensitive compared to? Try to be more precise and provide the p-value.

Response: More sensitive compared to S. aureus strains. The sentence was modified as follows: “Compared to S. aureus strains, Str. agalactiae and Str. dysgalactiae were more sensitive (P < 0.001) to the combination of the zinc pyrithione/gentamicin.”

Query: Line 109: Where are other graphs related to other bacterial strains tested?

Response: We added graph related to S. aureus strains (Figure 2), which were highly susceptible to the combination of antimicrobial agents at single concentration (0.25 µg/mL) of zinc pyrithione, to the Result section.

Query: Table 1-2-3: Please add the statistical differences using lowercase letters and p-values.

Response: We added the statistical differences using lowercase letters and p-values to the Tables 1, 2 and 3.

Query: Table 3: Substitute the comma with the dot for the separation of decimal numbers. Please list all values with two decimal places.

Response: In Table 3, we substituted the comma with the dot for the separation of decimal numbers and listed all values with two decimal places.

Query: Line 178: Causing the bacterial cell death.

Response: The sentence was modified as follows: “It has previously been documented that gentamicin is binding to the 30S ribosomal subunit, which leads to a misreading of the messenger ribonucleic acid (m-RNA), thereby inducing inhibition of protein biosynthesis, and causing the bacterial cell death [33].”

Query: Line 215: Even if the in vitro data can not be directly translated to in vivo, based on your results, which concentration of what molecules do you suggest for the treatment of skin bacterial infections?

Response: With aim to comment possible in vivo use of substances tested in our experiment, we added following text to the Discussion section: “However, our in vitro data cannot be directly interpreted for in vivo use, which requires the establishing of veterinary interpretive criteria such as pharmacokinetics and pharmacodynamics of tested compounds [47].”

Query: Line 256: Why did you measure at 405 nm of wavelength?

Response: The microtiter plates were measured at 405 nm of wavelength according to the methodology of Cos et al. The sentence was modified as follows: “The bacterial growth was then assessed as the turbidity determined by an Infinite 200 PRO microplate reader (Tecan, Männedorf, Switzerland) at 405 nm according to Cos et al. [50].”

Query: Statistical analysis is lacking, did you compare treatments and concentrations by statistical tests?

Response: We added following subsection 4.4. “Statistical analysis” to the Materials and Methods section: “The statistically significant differences (P < 0.001) between FIC values of the individual strains tested (in terms of the same concentration of zinc pyrithione within Tables 1, 2 and 3) were analysed using 2-way analysis of variance (ANOVA) with the general linear model procedure in SAS software (version 9.1, 2003, North Carolina, USA).”

Other changes

Numbers of references have been revised according to the manuscript changes.

Round 2

Reviewer 2 Report

The authors revised the manuscript according to my suggestions. The quality of the paper has significantly improved.

I have few minor comments before publication as follows:

Line 87-89: This sentence introduces the aim of the study, but it does not provide any hypothesis related to your research.

Line 237-240: Which concentration would you suggest for further testing? Can you provide your idea based on your experience with these compounds?

Lines 295-298: Remove the bold for this paragraph. Can you provide more information regarding the effect that were included in the model? Which post-hoc test did you perform for multiple comparisons? Statistically significance can be considered for P 0.05.

Author Response

Itemized response to the reviewer's comments

Manuscript-ID: antibiotics-1777487

Reviewer: 2

Query: Line 87-89: This sentence introduces the aim of the study, but it does not provide any hypothesis related to your research.

Response:.We added following hypothesis related with our research to the Introduction section: “Based on the results of our preliminary screenings performed as several combinations of zinc pyrithione with different antibiotics (erythromycin, clindamycin, gentamicin, oxacillin, penicillin, and vancomycin) against S. aureus of the American Type Culture Collection (ATCC) 29213, the combination of zinc pyrithione with gentamicin produced the lowest fractional inhibitory concentration (FIC) value 0.61 (unpublished data). Taking into account these laboratory results and into the synergistic potential of gentamicin against bacterial strains tested in this study [26,27], a theoretical growth-inhibitory synergistic effect of zinc pyrithione in combination with gentamicin could be hypothesized.”

Following references were added:

[26] Maia, N.L.; Barros, M.; Oliveira, L.L.; Cardoso, S.A.; Santos, M.H.; Pieri, F.A.; Ramalho, T.C.; Cunha, E.F.F.; Moreira, M.A.S. Synergism of plant compound with traditional antimicrobials against Streptococcus spp. isolated from bovine mastitis. Frontiers in Microbiology 2018, 9, 1–10 https://doi.org/10.3389/fmicb.2018.01203.

[27] Sreepian, A.; Popruk, S.; Nutalai, D.; Phutthanu, Ch.; Sreepian, P.M. Antibacterial activities and synergistic interaction of citrus essential oils and limonene with gentamicin against clinically isolated methicillin-resistant Staphylococcus aureus. The Scientific World Journal 2022, 1–12 https://doi.org/10.1155/2022/8418287.

Query: Line 237-240: Which concentration would you suggest for further testing? Can you provide your idea based on your experience with these compounds?

Response:. With aim to comment the concentrations recommended for further testing, we added following text to the Discussion section: “Although our in vitro data cannot be directly interpreted for in vivo use, 8-10 times the MIC of the substances tested, when the optimal bactericidal effect occurs, can be recommended for further research [49]. However, the MIC values gains clinical significance only in relation to pharmacokinetic parameters that describe the fate of the drug in the host organism, and therefore the next step in research should be their study [50].”

Following references were added:

[49] Maglio, D.; Nightingale, Ch.H.; Nicolau, D.P. Extended interval aminoglycoside dosing: from concept to clinic. International Journal of Antimicrobial Agents 2002, 19, 341–348 https://doi.org/10.1016/S0924-8579(02)00030-4.

[50] Kowalska-Krochmal, B.; Dudek-Wicher, R. The minimum inhibitory concentration of antibiotics: methods, interpretation, clinical relevance. Pathogens 2021, 10, 1-21 https://doi.org/10.3390/pathogens10020165.

Query: Lines 295-298: Remove the bold for this paragraph. Can you provide more information regarding the effect that were included in the model? Which post-hoc test did you perform for multiple comparisons? Statistically significance can be considered for P ≤ 0.05.

Response:. The bold for the subsection 4.4. “Statistical analysis” was removed. We would like thanks to reviewer, because based on its comment, we have found a typing error in this subsection (for statistical analyses 1-way ANOVA was used, not 2-way ANOVA). We provided more information about statistical analysis to this subsection. The text has been modified as follows: “The statistically significant differences (P < 0.001) between FIC values of the individual strains tested (in terms of the same concentration of zinc pyrithione within Tables 1, 2 and 3) were analysed using 1-way analysis of variance (ANOVA) with the general linear model procedure, followed by Tukey's HSD test, in SAS software (version 9.1, 2003, North Carolina, USA).5. The main effect was the individual bacterial strains.

Other changes

Numbers of references have been revised according to the manuscript changes.